# Person-Centered Climate, Garden Greenery and Well-Being among Nursing Home Residents: A Cross-Sectional Study

**DOI:** 10.3390/ijerph20010749

**Published:** 2022-12-31

**Authors:** Lijuan Xu, Yan Lou, Caifu Li, Xuemei Tao, Maria Engström

**Affiliations:** 1Medicine College, Lishui University, No. 1 Xueyuan Road, Lishui 323000, China; 2Faculty of Health and Occupational Studies, Department of Caring Science, University of Gävle, 801 76 Gävle, Sweden

**Keywords:** facility characteristics, greenery, nursing, nursing home, person-centered climate, staff, quality-of-care, well-being

## Abstract

Nursing home residents’ well-being is often proxy-rated in studies, and few studies have explored the association between resident-rated person-centered climate, garden greenery, and resident-rated well-being. A cross-sectional study was conducted. Questionnaire data from a convenient sample of 470 nursing home residents in a city in Southeast China in 2021 were analyzed using multiple linear regressions, with block-wise models. The instruments used were the Person-centered Climate Questionnaire-Patient version, the Nursing Home Greenery Index, and, for well-being, the EuroQol-Visual Analogue Scale, the Life Satisfaction Questionnaire, and the 9-item Patient Health Questionnaire (depression symptoms). In the unadjusted models, the person-centered climate was positively associated with general health (β 0.29, *p* < 0.001), person-centered climate and greenery with life satisfaction (β 0.39, and 0.18; both *p* < 0.001), and negatively with depression (β −0.28, and β −0.23, both *p* < 0.001). After adjusting for personal and nursing home characteristics, the associations between person-centered climate, greenery, and well-being remained statistically significant. The three models explained 36%, 35%, and 21% of the variance in general health, life satisfaction, and depression, respectively. This study provides knowledge on person-centered climate in long-term care and the access to greenery.

## 1. Introduction

According to WHO, long-term care for older people should integrate traditional health services, caregiving, and social support, all according to person-centered care principles [1]. Person-centered care, in turn, means engaging the person in the process of care [2], and the care should be delivered through the following activities: working with patients’ beliefs and values, engagement, having a sympathetic presence, sharing decision-making, and providing holistic care [3]. Another important aspect in the nursing home environment is the outdoor environment, the garden or patio, and the present study focuses on these two aspects, person-centered-care and the garden, in relation to resident-rated well-being.

In gerontological nursing, McCormack defined person-centeredness as being in relation, being in a social world, being in place, and being with self [4]. Person-centered care has been found to be related to nursing home residents’ quality of life [5,6], life satisfaction [7], and ability to perform activities of daily living (ADL) [8]. A systematic review and meta-analysis indicated that person-centered care could reduce depression and improve the quality of life for people with dementia [9]. Furthermore, a review of intervention studies in long-term care indicated that person-centered care improved residents’ psychological well-being [10], and an experimental study of individualized care found improved resident-rated person-centered climate, empowerment, and quality of everyday activities [11]. However, most of the mentioned studies [5,6,8] used proxy ratings, such as staff members’ ratings of person-centeredness and well-being for the residents, and only a few used resident-rated person-centeredness and resident-rated well-being.

Regarding greenery, there have been many studies conducted to explore the association between green space and health, e.g., [12,13,14,15,16]. Greenery contact, of any form or experience, will likely lead to positive health benefits. Studies within long-term care are fewer but have indicated that access to garden greenery in the nursing home is associated with an improvement in residents’ general health [17], quality of life [18,19], and mental well-being [20,21]. In addition, greenery view through the windows has also been associated with the improved mental health of nursing home residents [22]. A narrative systematic review revealed that garden use was positively associated with residents’ quality of life and negatively associated with depression [23]. Another review showed that, also, indoor nature exposure had positive physiological and psychological health benefits [24]. For greenery measurement, most of the mentioned studies used frequency of garden use [18,19] and duration of greenery exposure [20,21], whereas Dahlkvist et al. [17] measured the quantity and quality of greenery.

Based on the above-mentioned studies, the hypothesis was that the amount of person-centered care and garden greenery are positively associated with the well-being of the residents, and we examined the association, controlling for the residents’ personal characteristics and nursing home characteristics.

## 2. Materials and Methods

### 2.1. Design

A cross-sectional correlational study was conducted using the baseline data from a parental intervention study (training program for staff).

### 2.2. Setting and Participants

Using multistage, convenience sampling methods, we recruited 470 residents from 29 nursing homes in a Southeast Chinese city. The inclusion criteria specified residents who had lived in the nursing home ≥1 month and were able to respond to the questionnaire in a standardized, face-to-face survey. According to the recommendation from Polit and Beck [25], for multiple regression analysis, the sample size should be 50 + 8 times the number of determinants. Therefore, with the number of possible determinants at 20, there should be at least 210 study participants.

### 2.3. Data Collection

The data were collected from the end of July to October 2021. The resident data were collected by a trained research team, including the authors. The data regarding nursing home characteristics and the greenery index were collected from the nursing home managers or the regional government in a standardized, face-to-face, or phone questionnaire by the authors. The managers also identified the eligible residents who fulfilled the study’s inclusion criteria. Subsequently, the research team approached those residents interested in participating, in one big room of the nursing home, and informed them about the study. Informed consent was obtained from the residents who agreed to participate, and soap and towels were offered as gifts.

### 2.4. Measurement

Person-centeredness was measured using the Chinese version [26] of the 17-item Person-centered Climate Questionnaire-Patient version (PCQ-P) [27]. The scale includes three factors: safety (six items), everydayness (seven items), and hospitality (four items). The seven-grade response alternatives range from one (no, completely disagree) to seven (yes, completely agree). The factor scores are calculated using the mean score, and the scores for the total scale using the sum of the scores, ranging from 17 to 119. The PCQ-P Chinese version has demonstrated good validity and reliability (Cronbach’s alpha coefficient [α] = 0.93 total scale) among nursing home residents [26] and, in the present study, α = 0.96.

To measure greenery, the Nursing Home Greenery Index was used [17]. The first question concerns the natural features in the main view (response alternatives, including green areas, forest areas, and water areas). The second question is about the natural elements of greenery in the garden/patio (response alternatives include trees, shrubs, lawn, flowers, large stones, garden beds, kitchen gardens, raised garden beds, seasonal plants, bird baths, fountain/waterfall, ponds, and brooks). One point for each of the above elements yields one score. The total score of the greenery index is determined by the total number of natural views and natural elements of greenery, and the possible score range is from zero to sixteen. In addition, the question “what percentage of the residents visit the garden/patio during an ordinary day?” is posed. Greenery was assessed by the interaction between the greenery index and the percentage of residents visiting the garden/patio.

To measure well-being, the Life Satisfaction Questionnaire (LSQ) [28], the EuroQol-Visual Analogue Scale (EQ-VAS) [29], and the nine-item patient health questionnaire (PHQ) [30] were used. The Chinese version [31] of the 33-item LSQ was tested among the residents in nursing homes, revealing acceptable construct validity (using confirmatory factor analysis) and good reliability, with α ranging from 0.81 to 0.93 for the seven factors and total scale. The factors are physical symptoms (seven items, α present study = 0.69), sickness impact (six items, α = 0.82), quality of daily activities-fun (three items, α = 0.83), quality of daily activities-meaningful (four items, α = 0.87), socio-economic situation (three items, α = 0.77), quality of family relations (five items, α = 0.96), and quality of close-friend relationship (five items, α = 0.97). The seven-grade response alternatives range from one (strongly disagree) to seven (strongly agree). For the total score and factor scores, the corresponding items are summarized, divided by the highest possible factor score, and multiplied by 100. The total score of LSQ ranges from 14.3 to 100, a higher score indicating a better quality of life. EQ-VAS is an analog scale ranging from zero (worst imaginable health) to one hundred (best imaginable health) to indicate participants’ general health [29]. In addition, the EuroQol 5-Dimension-5 Level Questionnaire (EQ-5D-5L) [32] was used as the descriptive data for the sample (participants’ mobility, self-care, usual activities, pain/discomfort, and anxiety/depression). For each dimension, the five-grade response alternatives range from one (no problems) to five (unable to/extreme problems). To measure symptoms of depression, the Chinese version of the PHQ [33] was used. The response alternatives range from zero (not at all) to three (nearly every day). For the total score, the item scores are summarized, and higher scores indicate more symptoms of depression. The Chinese version of the PHQ has shown acceptable validity and reliability, α = 0.86 [34], and, in the present study, α = 0.80.

Personal characteristics (age, sex, marital status, education, having children, the main source of income, medical insurance, length of residence, family visits, multimorbidity, and frailty), and nursing home characteristics (number of residents, nursing home ownership, and types of rooms). To measure frailty, the seven-item FRAIL-Nursing Home (NH) Scale version 1 was used [35]. The three-grade response alternatives are zero (no problem, independent) to two (with a problem, dependent). The total score of the FRAIL-NH scale ranges from zero (best) to fourteen (worst). The Chinese version of FRAIL-NH has presented acceptable validity and reliability, with α = 0.67 [34], and, in the present study, α = 0.72. Staff and RN density were assessed using the ratio of the number of staff or RN/number of residents, which stands for the average number of staff or RN per resident in nursing homes.

### 2.5. Ethical Consideration

The study was approved by the medical ethics committee of XX University (No. 2021-0001). Before the data collection, all the participants received verbal and written study information, and those able to write signed an informed consent form. For the others, the researcher signed the form after verbal informed consent. The participants were informed that participation was strictly voluntary, that it would not affect their social support and healthcare in any way, and that they could drop out at any time during the study.

### 2.6. Data Analysis

The data were analyzed using IBM SPSS Statistics 26.0 (IBM Corp., Armonk, NY, USA). The internal missing data in the Life Satisfaction Questionnaire and the Person-centered Climate Questionnaire (a total of six points of missing data) were replaced with the resident’s median value for that factor. Descriptive statistics, such as the mean (standard deviation [SD]), median (25th and 75th percentiles inter-quartile range [IQR]), and percentage, were used to describe the sample. The Mann–Whitney U test was used to compare groups regarding well-being, and Spearman’s correlation coefficient for the bivariate correlation analyses. The effect size, r, for the Mann–Whitney U test was calculated using a standardized Z-score [36]. The variables with *p* ≤ 0.10, were included in the multiple linear regression models. In the first regression model (Model 1), person-centered climate was treated as a determinant. In the second model (Model 2), garden greenery was included as a determinant. The coefficient of determination (R^2^), the percent of variance explained by the model, is reported together with effect size. Cohen’s criteria [37] were used to determine effect size, f^2^ = R^2^/(1 − R^2^): 0.02 = small effect, 0.15 = medium effect, 0.35 = large effect. In the third model (Model 3), the covariates (personal characteristics and nursing home characteristics) were added. Personal characteristics that have been found associated with residents’ well-being (tested in the present study) were age, gender, e.g., [31,38,39,40], education [39,41], length of residence, e.g., [31,38,40,42], family visits [38,39], medical insurance [31], chronic disease [41,42], and frailty [43,44]. In addition, facility characteristics that have been found associated with residents’ well-being include nursing home ownership [31,39], staff and registered nurse (RN) density, e.g., [45,46,47], types of rooms [39], and the number of residents [45]. *p* < 0.05 was considered statistically significant.

## 3. Results

Among the included nursing homes, there was one public, 15 public–private, and 13 private nursing homes. The number of residents varied widely, ranging from 20 to 328. The median (IQR) ratio of RNs/residents was 0.004 (0:0.03) and staff/residents was 0.23 (0.18:0.30). The mean age of the participants was 82.7 years (SD = 8.3), and 48% were women. The median (IQR) length of residence was two (1.0:4.0) years. About half (58%) of the participants resided in public or public–private nursing homes (Table 1).

### 3.1. Participant-Nursing Home Characteristics and Well-Being

Table 2 and Table 3 indicate that the variables: age, education, length of residence, family visits, sex, having children, the main source of income, medical insurance, multimorbidity, frailty, nursing home ownership, ratio of RNs/residents, ratio of staff/residents and room types, had *p*-values ≤ 0.10 in the analyses of well-being (general health, life satisfaction, or depression), and these variables were adjusted in the multiple regression analyses (except having children and ratio of total staff/residents, which were excluded due to multicollinearity with family visits and the ratio of RNs/residents).

### 3.2. Bivariate Associations between Person-Centered Climate, Greenery, and Well-Being

Garden greenery was positively associated with life satisfaction (rs 0.17), and negatively with depression (fewer symptoms of depression, rs −0.21). Person-centered climate was positively associated with general health (rs 0.30) and life satisfaction (rs 0.38), and negatively with depression (rs −0.28) (Table 3).

### 3.3. Multiple Linear Regression Models

In the unadjusted Models 1 and 2, the standardized coefficient of person-centered climate on general health was 0.29 (*p* < 0.001). The standardized coefficient for person-centered climate and greenery on life satisfaction was 0.39 and 0.18 (all *p* < 0.001). The standardized coefficient of person-centered climate on depression was −0.28 and greenery −0.23 (both *p* < 0.001). The explained variance was 9% in general health, 20% in life satisfaction, and 15% in depression. After adjusting for personal and nursing home characteristics, the associations between person-centered climate and greenery with well-being remained statistically significant; the total variance explained by all the variables in the models was 36% for general health (effect size medium [f^2^ = 0.15]), 35% for life satisfaction (effect size small [f^2^ = 0.14]), and 21% for depression (effect size small [f^2^ = 0.05]) (Table 4).

## 4. Discussion

To our knowledge, this is the first study to explore the association between person-centered climate, garden greenery, and nursing home residents’ well-being. The adjusted multiple regression models indicated that a person-centered climate is positively associated with general health and life satisfaction and negatively associated with depression, and greenery is positively associated with life satisfaction and negatively associated with depression.

Our findings of the association between resident-rated person-centered climate and well-being are supported by previous research [5,6] and are in line with WHO recommendations [1]. However, previous studies have mostly used proxy measures (from staff) for well-being and person-centered care. Compared with staff, Yang et al. [26] found that residents scored lower on person-centered climate than staff. In a previous study [7], it was indicated that, among the dimensions of person-centered climate, safety and everydayness were positively associated with life satisfaction among nursing home residents. In line with our hypothesis, we also found that access to garden greenery (greenery index * percentage of garden visits) was associated with increased life satisfaction and decreased depression. The result is consistent with previous studies and indicates the importance of access to a garden at the nursing home with green spaces. In a longitudinal study [48], it was found that views of the greenery outside the nursing home were associated with less stress and increased quality of life among the residents. A study conducted in six European countries revealed that garden greenery was beneficial for nursing home residents’ physical activities, recreation, social interactions, and quality of life [19]. The pathways linking greenery and health could include air quality, physical activity, restoration, asocial cohesion, and stress reduction [49]. A study of nursing home residents in Sweden also indicated that garden greenery had an indirect effect on health, mediated by restorative variables such as being away and fascination [17]. Our findings showed that access to garden greenery was associated with life satisfaction and depression. The data were collected during the COVID-19 pandemic, and the nursing home residents were usually socially isolated, facing restrictions on family visiting [50] and limitations on social activities [51] during this time period. During the COVID-19 pandemic, for nursing home residents, access to garden greenery seemed to be an important choice for social participation, which could decrease emotional distress [52]. In addition, our findings also indicated that the greenery index alone (i.e., not in combination with average visits; see Appendix A) was associated with well-being, and the results support the perspectives that a nature view through a window may also influence health [53]. However, the regression coefficient for greenery was weaker when it was not combined with the average percentage of visits to the garden (Model 3, Appendix A, standardized coefficient 0.10 (vs. 0.18, Table 4) for life satisfaction and −0.14 (vs. −0.25, Table 4) for depression. Regarding distant window views, Kent et al. [54] found a difference in visual satisfaction levels depending on whether the nearby landscape contained nature or urban features. In the present study, the surrounding conditions, such as forests or other natural landscapes, were not included in the greenery index if not seen from the garden/patio. Among the 29 nursing homes, only 2 nursing homes in the study were in the countryside, surrounded by forest or farms, while the others were in a city center or surrounded by village houses. Our results are consistent with the findings of Brook et al. [55], where outdoor nature contact, and also pictured nature, decreased negative moods (depression, stress, or anxiety), but nature contact had s stronger impact. The access to, amount of, and types of greenery should be an important consideration for nursing homes’ garden design. Staff supporting resident outdoor visits, including those using wheelchairs to go outside, should be encouraged by the managers. The models tested in the present study explained 36% (*p* < 0.001), 35% (*p* < 0.001), and 21% (*p* < 0.001) of the variance in general health, life satisfaction, and depression, respectively (effect sizes small to medium). The *p*-value is influenced by two parameters: the effect of interest, and the sample size. The effect size is an important parameter, particularly if the sample size itself is quite large [36]. Although the models explained less than the desired 50% of the variance, it was more than in the study by Dahlkvist et al. [17], where 17.6% of the variance in resident-rated health was explained by garden greenery, resident-rated being away, fascination, and visits in the garden. It is important to remember, also, that the greenery index is a unit-level variable, assessed by the manager, and not the residents’ perception of the garden, so the expected variance might thus be weaker. On the other hand, the risk of common-method variance is lower. Thus, even if our results are statistically significant, the clinical significance of the results could be questioned, due to the mostly small effect sizes, and the results need to be interpreted together with results from other studies with the same focus.

Finally, consistent with previous studies [45,56], our findings revealed that the covariate ratio of RN/residents (and staff/residents) was associated with life satisfaction. According to Røen et al. [46], higher staff density might mean more time for developing personal relationships with residents, more personalized activities, and a higher level of person-centered care for the nursing home residents. However, in our results, staff and RN density was not associated with a person-centered climate. Nevertheless, it was associated with life satisfaction. To integrate medical care (c.f. WHO recommendations for long-term care services), our findings support the need for more RN/residents to improve resident life satisfaction.

### Methodological Consideration

This study has several limitations. First, the cross-sectional design limits cause-and-effect links; and second, only about 16% of the participants had moderate or severe problems with mobility, self-care, or usual activities, whereas, in other studies [57,58], the proportion of residents with physical functional impairment in nursing homes was about 50%. However, we adjusted for frailty in the multiple regression models. Third, during the data collection, the managers identified the eligible residents, and there may have been some residents who declined and were not mentioned by the managers. The inclusion criterion “being able to respond to the questionnaire” was not assessed with any tests for cognitive function. The estimation of garden greenery was conducted by the nursing home managers and not by the researchers, allowing bias in the overestimation of greenery and the percentage of garden visits, and, in the future, it would be interesting to add the residents’ perception of ‘being away’ and ‘fascination‘ when visiting the garden, as in the study by Dahlkvist et al. [17]. In addition, we did not accurately gauge the total amount of greenery (e.g., garden size, number of windows, etc.). The Chinese version of the person-centered climate questionnaire indicated multicollinearity among the subscales, and thus we used the total scale in the multiple regression analyses. The statistically significant associations between greenery and person-centered climate, and the outcomes, are mostly small (e.g., rs 0.17 to 0.38, Table 3) due to the large sample size, and the statistically significant differences in well-being for the subgroups (Table 2) all have small effect sizes. For the regression models, the effect sizes range from small (depression and life satisfaction) to medium (general health). The strengths of the study are that the models were adjusted for personal and nursing home characteristics; there were few missing data points; and residents’ own ratings of person-centered climate and well-being were used compared to the earlier use of proxy ratings. On the other hand, by using resident-rated person-centered care climate and well-being rather than proxy-rated, several residents could not be included, as they did not fulfill the study’s inclusion criteria of being able to respond to the questionnaire in a structured interview.

## 5. Conclusions

Our findings indicate that a person-centered climate and access to garden greenery may contribute to improved residents’ well-being.

## Figures and Tables

**Table 1 ijerph-20-00749-t001:** Personal characteristics of residents and nursing home characteristics (*n* = 470).

Variables	*n* (%)	Mean (SD)/Median (IQR), Min–Max
Age (years)		82.7 (8.3), 52–101
Length of residence (years)		2.0 (1: 4), 0.1–25
Times of family visits ^a^		3 (0: 5), 0–6
Sex		
Female	228 (48.5)	
Male	242 (51.5)	
Marital status		
Divorced/single/widow(er)	405 (86.2)	
Married	65 (13.8)	
Education		
No formal/primary school	359 (76.4)	
Junior high school	72 (15.3)	
High school	26 (5.5)	
College or higher	13 (2.8)	
Having children		
No	118 (25.1)	
Yes	352 (74.9)	
Main source of income		
Family members or others	310 (66.0)	
Retirement pension	160 (34.0)	
Medical insurance		
Medical insurance for employees	85 (18.1)	
Basic medical insurance for residents or self-paid	385 (81.9)	
Multimorbidity		
No	261 (55.5)	
Yes	209 (44.5)	
Number of residents/nursing home		85 (46: 200), 20–328
Ratio of RNs/residents		0.004 (0:0.03), 0–0.09
Ratio of total staff/residents		0.23 (0.2:0.3), 0.12–0.46
Greenery index		10 (9:11), 6–13
Percentage of residents visiting garden		0.6 (0.5:0.8), 0.1–1.0
Nursing home ownership		
Public and public–private nursing home	275 (58.5)	
Private nursing home	195 (41.5)	
Type of room (number of residents sharing the same bedroom)		
Double or more	286 (60.9)	
Single room or a room for couples	184 (39.1)	
EQ-5D-5L		
Mobility (no problems/slight problems/moderate problems/severe problems/unable)	283/108/35/16/28	
Self-care (no problems/slight problems/moderate problems/severe problems/unable)	356/61/21/8/24	
Usual activities (no problems/slight problems/moderate problems/severe problems/unable)	337/79/28/12/14	
Pain/discomfort(no/slight/moderate/severe/extreme)	220/160/60/25/5	
Anxiety/depression(no/slight/moderate/severe/extreme)	356/80/20/11/3	

Notes: SD Standard deviation, IQR Inter-quartile range, RNs Registered Nurses. ^a^ The ordinal variable and the alternative responses are 0 never, 1 less than once a month, 2 once a month, 3 several times a month, 4 once a week, 5 several times a week, 6 every day.

**Table 2 ijerph-20-00749-t002:** Well-being compared for different subgroups (*n* = 470).

Variables Mean (SD)	EQ-VAS ^c^	LSQ ^c^	Depression ^d^
Sex, effect size (*p*-value) ^a^	0.043 (0.356)	0.098 (0.034)	0.055 (0.229)
Female	70.1 (16.9)	66.8 (10.9)	2.8 (3.8)
Male	70.7 (20.1)	64.6 (10.4)	2.4 (3.4)
Marital status, effect size (*p*-value) ^a^	0.004 (0.937)	0.072 (0.121)	0.011 (0.805)
Divorced/single/widow(er)	70.6 (18.1)	65.4 (10.7)	2.6 (3.6)
Married	69.2 (21.5)	67.6 (10.7)	2.9 (4.2)
Having children, effect size (*p*-value) ^a^	0.021 (0.644)	0.279 (<0.001)	0.035 (0.448)
No	70.3 (21.1)	60.3 (10.4)	2.9 (3.8)
Yes	70.4 (17.7)	67.5 (10.2)	2.5 (3.6)
Main source of income, effect size (*p*-value) ^a^	0.107 (0.020)	0.132 (0.004)	0.006 (0.898)
Family members or others	68.6 (19.9)	64.6 (10.9)	2.8 (3.9)
Retirement pension	73.7 (15.1)	67.7 (10.0)	2.3 (3.0)
Medical insurance, effect size (*p*-value) ^a^	0.046 (0.314)	0.147 (0.001)	0.061 (0.185)
Medical insurance for employees	71.8 (18.7)	68.7 (10.1)	2.7 (3.5)
Basic medical insurance for residents or self-paid	70.1 (18.6)	65.0 (10.7)	2.6 (3.7)
Multimorbidity, effect size (*p*-value) ^a^	0.294 (<0.001)	0.126 (0.006)	0.225 (<0.001)
No	75.3 (16.5)	67.0 (10.0)	1.9 (2.8)
Yes	64.2 (19.2)	64.0 (11.3)	3.5 (4.3)
Nursing home ownership, effect size (*p*-value) ^a^	0.145 (0.002)	0.001 (0.975)	0.055 (0.236)
Public and public–private nursing home	72.5 (18.3)	65.8 (10.7)	2.3 (3.2)
Private nursing home	67.3 (18.6)	65.5 (10.6)	3.0 (4.2)
Number of residents in nursing home, effect size (*p*-value) ^a,b^	0.007 (0.873)	0.031 (0.508)	0.030 (0.512)
Fewer residents (≤85 residents)	69.9 (20.3)	65.9 (11.3)	2.8 (4.0)
More residents (>85 residents)	70.8 (16.5)	65.5 (10.1)	2.3 (3.2)
Type of room (number of residents sharing the same bedroom), effect size (*p*-value) ^a^	0.152 (0.001)	0.116 (0.012)	0.109 (0.018)
Double or more	68.0 (19.9)	64.7 (11.2)	2.9 (3.9)
Single or couple	74.0 (15.6)	67.2 (9.7)	2.1 (3.1)

Note: ^a^ effect size and *p*-value for Mann–Whitney U test, effect size calculated using standardized Z-scores [36], 0.1 = small effect, 0.3 = medium effect, 0.5 = large effect [37]; ^b^ groups divided by median; ^c^ a higher score is more desirable; ^d^ a lower score is more desirable. EQ-VAS EuroQol-Visual Analogue Scale, LSQ life satisfaction questionnaire.

**Table 3 ijerph-20-00749-t003:** Relationships between study variables.

Variables	Mean (SD)/Median (Q1:Q3)	(1)	(2)	(3)	(4)	(5)	(6)	(7)	(8)	(9)	(10)	(11)	(12)	(13)	(14)	(15)
Age (1)	82.7 (8.3)	--														
Education (2)		0.09	--													
Length of residence (3)	2.0 (1: 4)	−0.12 *	−0.13 **	--												
Family visits (4)	3 (0: 5)	0.51 ***	0.22 ***	.−0.47 ***	--											
Ratio of RNs/residents (5)	0.004 (0:0.03)	0.33 ***	0.28 ***	−0.25 ***	0.51 ***											
Ratio of total staff/residents (6)	0.23 (0.2:0.3)	0.31 ***	0.28 ***	−0.27 ***	0.45 ***	0.72 ***										
Greenery index×percentage visit of garden (7) ^a^	6.0 (2.4)	−0.03	−0.14 **	0.04	−0.03	−0.22 ***	−0.32 ***									
Safety (8)	5.3 (1.1)	0.09 *	0.09	−0.07	0.11 *	0.04	−0.01	0.08	--							
Everydayness (9)	5.5 (1.0)	0.05	0.12 *	−0.01	0.06	0.02	−0.04	0.09	0.89 ***	--						
Hospitality (10)	5.2 (1.1)	0.01	0.06	−0.01	0.03	−0.05	−0.08	0.12 **	0.82 ***	0.84 ***	--					
Person-centered climate total (11)	91.0 (16.8)	0.06	0.10 *	−0.03	0.08	0.02	−0.04	0.10 *	0.96 ***	0.96 ***	0.91 ***	--				
Frailty (12)	1.5 (2.0)	0.02	−0.02	−0.12 **	0.11 *	0.07	0.09 *	−0.16 **	−0.20 ***	−0.24 ***	−0.25 ***	−0.24 ***	--			
EQ-VAS (13)	70.4 (18.6)	0.07	0.16 ***	−0.01	0.05	−0.01	−0.06	0.05	0.29 ***	0.30 ***	0.26 ***	0.30 ***	−0.47 ***	--		
LSQ (14)	65.7 (10.7)	0.20 ***	0.07	−0.18 ***	0.23 ***	0.15 **	0.13 **	0.17 ***	0.36 ***	0.36 ***	0.36 ***	0.38 ***	−0.34 ***	0.41 ***	--	
Depression (15)	2.6 (3.6)	−0.00	0.02	0.09 *	−0.07	0.04	0.05	−0.21 ***	−0.26 ***	−0.25 ***	−0.32 ***	−0.28 ***	0.52 ***	−0.40 ***	−0.50 ***	--

Note: values from Spearman correlation analysis. -- The variable was not included in the model. SD Standard deviation, RNs registered nurses, EQ-VAS EuroQol-Visual Analogue Scale, LSQ life satisfaction questionnaire, * *p* < 0.05, ** *p* < 0.01, *** *p* < 0.001. The correlation, rs, between safety, everydayness, and hospitality greater than 0.8, indicating multicollinearity. ^a^ Association between greenery index with well-being in the Appendix A.

**Table 4 ijerph-20-00749-t004:** Linear regression analyses with well-being as the dependent variable and person-centered climate and greenery as independent variables (Models 1–2), adjusted for personal characteristics and NH characteristics (model 3) ^a,b^ (*n* = 470).

	EQ-VAS	LSQ	Depression ^c^
**Unadjusted Model 1, R square (Adjusted R Square) *p* value**	0.09 (0.08), <0.001	0.17 (0.16), <0.001	0.10 (0.09), <0.001
Person-centered climate	0.29 ***	0.41 ***	−0.31 ***
**Unadjusted Model 2, R square (Adjusted R Square) *p* value**	0.09 (0.08), <0.001	0.20 (0.20), <0.001	0.15 (0.14), <0.001
Person-centered climate	0.29 ***	0.39 ***	−0.28 ***
Greenery index × percentage of visits garden	--	0.18 ***	−0.23 ***
**Adjusted Model 3, R square (Adjusted R Square) *p* value**	0.36 (0.35), <0.001	0.35 (0.34), <0.001	0.21 (0.20), <0.001
Person-centered climate	0.20 ***	0.32 ***	−0.27 ***
Greenery index × percentage of visits garden	--	0.18 ***	−0.25 ***
Frailty	−0.42 ***	−0.26 ***	-- c
Age	--	0.09 *	--
Education	0.08 *	--	--
Length of residence	--	−0.10 *	0.04
Family visits	--	0.09	--
Sex (male = 1, female = 0)	--	−0.05	--
Main source of income (retirement pension = 1, others = 0)	0.10 *	−0.02	--
Medical insurance (basic medical insurance for residents and self payment = 1, Medical insurance for employment = 0)	--	−0.09 *	--
Multimorbidity (yes = 1, no = 0)	−0.22 ***	−0.11 **	0.22 ***
Nursing home ownership (private nursing home = 1, public and public–private nursing home = 0)	−0.01	--	--
Room type (single or couple room = 1, double or more = 0)	0.03	−0.04	−0.09 *
Ratio of RNs/residents	--	0.12 *	--

Note: Values of standardized regression coefficient. -- The variable was not included in the model. ^a^ When sub-scales of person-centered climate were entered in the regression models, variance inflation factors (VIF) were more than 2.5, indicating multicollinearity, and therefore total score of person-centered climate was included in the analyses. ^b^ When having children and ratio of total staff/residents were entered in the regression models, VIF were more than 2.5, indicating multicollinearity, therefore having children and ratio of total staff/residents were excluded in the analysis (leaving family visits and ratio of RNs/residents in the analysis). ^c^ One item of frailty scale is the same as depression, and therefore Frailty scale was not included in the analysis for Depression. Abbreviations: EQ-VAS EuroQol-Visual Analogue Scale, LSQ life satisfaction questionnaire, RNs registered nurses, NH nursing home. * *p* < 0.05, ** *p* < 0.01, *** *p* < 0.001.

## Data Availability

The data that support the findings of this study are available on request from the corresponding author. The data are not publicly available due to privacy or ethical restrictions. Therefore, they are available from the corresponding author upon reasonable request.

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
