# Peer review of "Person-Centered Climate, Garden Greenery and Well-Being among Nursing Home Residents: A Cross-Sectional Study"

_ijerph, 2022, doi:10.3390/ijerph20010749_

Round 1
Reviewer 1 Report
I am a statistician who has supported researchers with similar questions. I focused my review on the statistical aspects of your submitted paper and offer the following comments:
1. Line 61. Hypotheses should be stated in terms of constructs rather than measures of constructs. I suggest deleting 'higher scores'.
2. Line 74. Good rationale for sufficient sample size.
3. Lines 89-130. I am not familiar with the measures. The other reviewers should justify their usage.
4. Lines 79 and 142 share the same content.
5. Line 149. SPSS should be referenced.
6. Table 3 reports a number of small correlations that are statistically significant due to sample size.
7. Line 200-208. Again, statistical significance is due to the large sample size. What would be a meaningful magnitude of an R-square in this area of research?
8. Table 4. Adjusted R-squares are less than 0.50; a desired level. These results should be rationalized in the Limitations section.
9. Statistically, there are a number of significant results that are a result of the large sample size. Attention should be paid to the clinical significance of the findings. If I was a nursing home manager and had to decide whether to begin a Garden Greenery project or something else, have you convinced me that the Garden Greenery is effective for the well-being of my residents?
Reviewer 2 Report
Dear authors,
please complete the survey with direct user evaluation. It is also worth referring to a greater extent to foreign research in other cultures and correlating them with the results achieved.
Best regards
Reviewer
Reviewer 3 Report
Thank you for preparing this interesting paper. I thought that the topic was of high interested, and the results generally supported what has been found in the literature for other contexts. An area of limitation seemed to be that little information was known about the existing building and surrounding conditions to fully understand the role nature had in this study. I have highlighted this, in detail, in my comments below, and thought that if the authors could acknowledge these remarks this could help improve their work.
1) Intro: The introduction is generally well structured. I was wondering whether the provision for the outdoor environment is primarily to connect elderly persons to the outside, and/or so they have direct access immersive to these spaces. This is briefly mentioned on p2, line 54, but was not elaborated in much detail, besides some lines (242-243) that were in the discussion. I would assume that it is both. Hence, it may be useful to also mention similar benefits nature creates here; for example, through window views (e.g., O’Connor et al, 1991. Window view, social exposure and nursing home adaptation) and from indoor plants (e.g., Mcsweeney et al, 2014. Indoor nature exposure (INE): a health-promotion framework.)
2) Method: Similar to my previous comment; the method contained a question asking the percentage of residents visit the garden/patio on an ordinary day (p3, lines 105-107.) This could mean that the “Greenery Index” could have had a correlative influence with other means of nature connection. If the authors have any additional information about the buildings and gardens they had surveyed (e.g., size and location, number of windows, garden size, etc.), it would be useful to include them in the method.
3) Method: On page 3, lines 99-102, a checklist for nature items present within the garden/patio was provided. A problem with this appropriate is that not enough information is known about the surrounding building site, to accurately gauge the total amount of greenery. For example, if the building is located within a city-center, this method may accurately represent the total amount of nature. However, if buildings are surrounded by forests or other sources of nature (e.g., distance natural landscape), will this be a reliable measure for nature? I think this would make an interesting discussion point.
4) Data analysis: Please explain which groups the Mann-Whitney U tests compared (line 153.) I could not find this information, and it was not clear what Table 2 showed. Since the sample size is also relatively large, please consider also including the effect sizes (e.g., Cliff’s Delta, or Pearson’s r) for the Mann-Whitney U tests.
5) Table 1: I thought that the values for this table could have been presented better for the readers. They could have been separated into their own columns with column headers labeled to indicate which value they represented, and how they were expressed (e.g., as a percentage or a type of average.)
6) Table 3: Please consider presenting this information as a graphical matrix, with colors that can be used to indicate the strength of association between the variables. Currently, there is too much information for the readers to understand when these results are presented in a tabular format. Also, the second column (averages) can be removed, as these do not add much to the plot.
7) Discussion: Page 7, line 242: Thank you considering the influence of window view here. I thought there were some considerations that the authors could include here. Nature that is closer to the window from the outside (also related to comment #3) will generally higher positive influence when seen through the view (Kent et al, 2020. Evaluation of the effect of landscape distance seen in window views on visual satisfaction.) I am also wondering whether surrounding site conditions, beyond the garden, were also important to this study. Views of nature in distant landscapes would likely have been present for some buildings, and a discussion on this would have been useful. Also, when people are immersed in the garden, the type of nature contact (e.g. through a view or immersive) could have also influenced the health benefits people received (Brook et al, 2017. Nature-related mood effects: Season and type of nature contact) I would assume that these are larger when they are in their garden, rather than look at them through the window view.
8) Season was also measured (P3, line 101), so a brief discussion on seasonal influences could also be included.
Round 2
Reviewer 3 Report
Thank you for considering all my previous comments. I think this has made a significant improvement to this paper. I enjoyed reading through the revised work, and would like to put forward some final, yet relatively minor suggestions, for further consideration:
#1. Page 2: It would be useful to briefly mention that nature contact, of any form or experience, will likely lead to positive benefits. This can then lead into the different types of contact (e.g. views and gardens) people are likely going to experience in care homes.
#2. Page 4, line 160: "R-squared" please change this to "The coefficient of determination." Also, thank you for providing the effect size thresholds. This was appreciated.
#3. Table 1: Thank you for reformatting the table. I think this looked much improved. For the Q1:Q3, I was wondering whether this is the 25th and 75th quartiles? If so, perhaps this could be stated as the IQR (inter-quartile range.)
#4. Page 9, line 287-289: The interpretation of the effect size along with p-value was valuable. However, the authors may also want to state that the p-value is influenced by two parameters: the effect of interest, and the sample size. This sometimes makes the effect size an important parameter, particularly if the sample size itself is quite large. Interpreting this two parameters in the context of this research endeavor may help readers better understand the results.
